# Laboratory Testing and Theoretical Modeling of Deformations of Reinforced Soil Wall

**Krystyna Kazimierowicz-Frankowska * and Marek Kulczykowski**

Department of Geomechanics, Institute of Hydro-Engineering of Polish Academy of Sciences, ul. Kościerska 7, 80-328 Gdansk, Poland; marek@ibwpan.gda.pl
* Correspondence: krystyna@ibwpan.gda.pl; Tel.: +48-58-552-29-62

**Abstract:** This paper presents the results of an experimental investigation of a vertical reinforced soil (RS) wall. The structure was built on a laboratory scale. Horizontal displacements on the surface of the model wall were monitored at the end of construction and during surcharge application (as post-construction displacements). The experimental results were compared with their theoretical predictions. The accuracy of the selected analytical approach was examined to predict deformations of the RS structure under external loading. It was shown that the proposed original and relatively simple analytical method for estimating structural deformation can be successfully used in practice (the average difference between the recorded and calculated values of deformation did not exceed 25%). From a scientific point of view, an important element of this work was the analysis of the effect of friction between the backfill and the side walls of the test box on the measured displacements. For the investigated case, it was shown that the impact of this element caused a reduction in the value of external loading of more than 60%. The final results may be particularly useful in the design process of structures used in transportation engineering (bridge abutments), where deformation limit values cannot be exceeded.

**Keywords:** reinforced soil walls; experimental tests; facing displacements; analytical methods





## 1. Introduction

The use of reinforced soil (RS) walls, treated as a new, more economical proposition compared to conventional retaining walls, has increased significantly across the world during the last few decades. In recent periods of time, a large number of such structures has been built because of their numerous advantages, including reliability, aesthetics, cost effectiveness, simplicity of construction, tolerance of differential settlements, and good seismic performance [1–13].

Predicting the behavior of reinforced soil walls, especially under external loading, is complicated by the varied properties of their component materials (such as soil, reinforcement, wall elements), the complexity of their immediate interactions, different boundary conditions, variable structural geometry, and the different construction methods used in particular cases. Several investigations have been performed in order to recognize how the behavior of reinforced soil walls is influenced by particular design factors, such as wall geometry, the types of loading, foundation quality, the facing type and its connection with reinforcement, the reinforcement type and tensile parameters, the reinforcement layout in horizontal and vertical directions, and backfill parameters.

The available results show that the effectiveness of RS wall technology is significantly dependent on the interaction between two essential materials used to build this type of structure: the soil that is used as the backfill and the reinforcements. In a correctly constructed RS wall, this interaction substantially increases the tensile strength inside the soil mass, allowing the overall structure to behave in a way similar to that of a monolithic body, supporting not only its own weight, but also external loads [14–18].

To improve knowledge in this area, many researchers have studied both the static performance of reinforced soil walls [19–26] and the dynamic behavior of such structures. They have performed both experimental tests and theoretical modeling. The analysis of typical RS walls is still often carried out using analytical methods, but in many (especially more complex) cases, a numerical simulation by FEM may also be useful and is included in the design process. In addition, it is often worth comparing the results obtained using different methods. The most commonly used approach takes into account the assumption that the plane strain conditions and calculations are carried out in a two-dimensional (2D) framework [27–35].

The different design concepts for RS walls using particular types of materials as reinforcement have been extensively analyzed and clearly described in the literature [36–38]. However, it should be noted that the methods of designing such structures have focused mainly on their ultimate limit states, such as pull-out and global and local stability failure. Although many studies have investigated the influence of selected reinforcement parameters on the behavior of reinforced soil foundations, research on the serviceability limit state has been scarce [39,40]. Deformation in RS structures is becoming an ever more important design consideration, as such structures are built with increasingly tight tolerances. It is crucial, for example, when they are used as elements of infrastructure in road engineering. The performance of such structures as geosynthetic reinforced soil (GRS) bridge abutments under service loads and working stress has been reported as satisfactory from engineering point of view [41–49]. That is why the problem of deformation in such structures needs further investigation.

The main aim of this study is to present the results of our experimental investigations into deformations in a reinforced soil wall, and to compare these results with the theoretical predictions of these deformations. For this purpose, we used an analytical method involving the calculation of displacements of reinforced soil structures. The method, elaborated by [40], is an extension of a method proposed by [18,50,51].

The final results may be particularly useful in the design process of structures used in transportation engineering (bridge abutments), where deformation limit values cannot be exceeded.

## 2. Methods

### 2.1. Background

Several studies have dealt with the behavior of footings constructed on stabilized sand walls [52–70]. Some authors are interested in both experimental and theoretical investigations of RS structures. For example, Adams et al. [71] built a few large-scale mini-pier tests using silty gravel as back-fill material, concrete masonry units as facing elements, and woven geotextiles of different tensile strengths and vertical spacing as reinforcement layers. The authors concluded that the performance of the RS mass and its behavior under loading was more dependent on reinforcement spacing than on geotextile strength. Pham [72] carried out large-scale generic soil-geosynthetic composite tests under plane strain conditions. The gravel was used as back-fill material and woven geotextiles of different tensile strengths and vertical spacing were used as reinforcement layers. During the experiments, Pham [72] applied different confining pressures to the two sides of the reinforced soil mass. The author [72] pointed that vertical spacing of the reinforcement had a more significant influence on the performance of the RS mass than its tensile strength.

Based on published results [72,73], it can be concluded that large-scale loading tests are very useful to investigate the real behavior of RS structures, considering their aggregate size and reinforcement spacing [74,75]. However, it is worth bearing in mind that this kind of investigation also has some disadvantages. Nicks et al. [76] paid attention to two major problematic items associated with large-scale RS performance tests. Firstly, these tests require specialized equipment which is not always available. Secondly, a performance test gives results only for a specific case (for a specific RS structure model characterized by an individually selected combination of parameters). Therefore, the test results obtained

cannot be considered representative of other structures—without additional checking—when the geometry and/or materials will be changed. Additionally, it should be noted that this is a very expensive type of research that requires significant funding. Moreover, in some cases, many interesting results can be obtained by limiting activities to small-scale experiments. This is the approach that was used during this research, the results of which are presented in the article.

In this study, special attention was paid to the theoretical analysis of the behavior of RS walls under serviceable loading. The main part of this investigation focused on recognizing the scope and character of these deformations. It should be noted that only limited studies concerning this aspect of the behavior of RS structures are presented in the literature. However, the deformation of RS walls is becoming an ever more important design consideration since they are built with increasingly tight tolerances. The deformation performance of these composite structures, which combine the benefits of strong soil and reinforcement, can be influenced by a large number of factors. The most important are the geometrical properties of RS structures, the time-dependent characteristics of the reinforcement, and the main parameters of the soil used as backfill and subgrade/foundation materials.

During their professional activities, engineers are interested mainly in accurately predicting two types of deformations of RS structures: horizontal displacements of the RS facing and vertical displacements of the crest (Figure 1). The scope of this paper is limited to the analysis the former. It should be noted that additional information regarding the problems associated with the deformation of RS structures is available from other published articles [50,51,77–80].

There are two general mathematical approaches that can be used to solve the problem. The first one is based on analytical models for the calculation of deformations of RS structures. This approach depends on accurate and precise assumptions regarding the workings of RS structures and their particular elements. Modeling errors cause significant inaccuracies in the output results. Thus, such methods require good theoretical background regarding the mechanisms of deformations in RS structures in order to be useful.

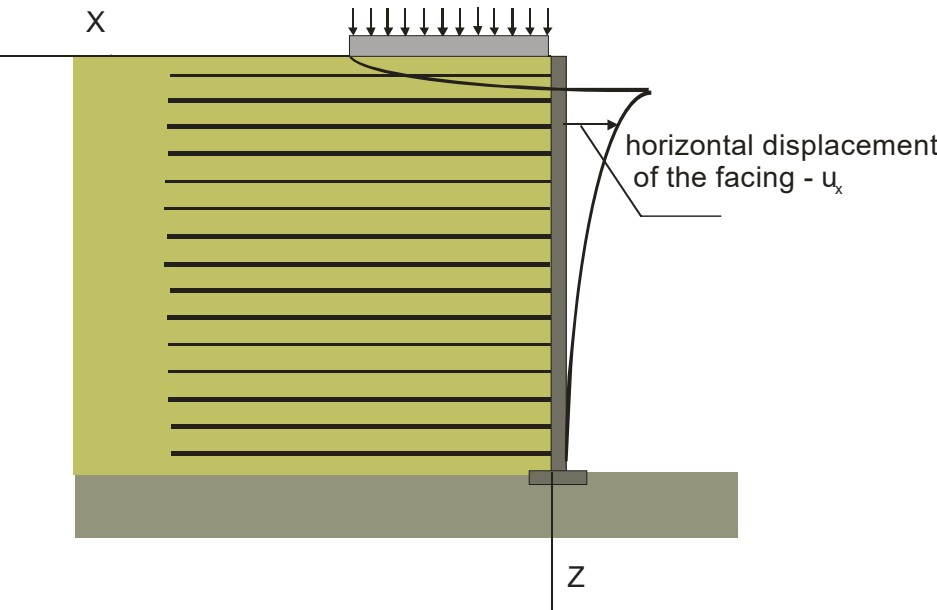

**Figure 1.** Deformations of RS structure under serviceable loading.

Although the number of studies has increased, so far there is no single accepted method which could be used in the design process to predict deformations in RS structures. The calculation methods that are available can be classified into a few groups (Figure 2), all of which have advantages and disadvantages [51].

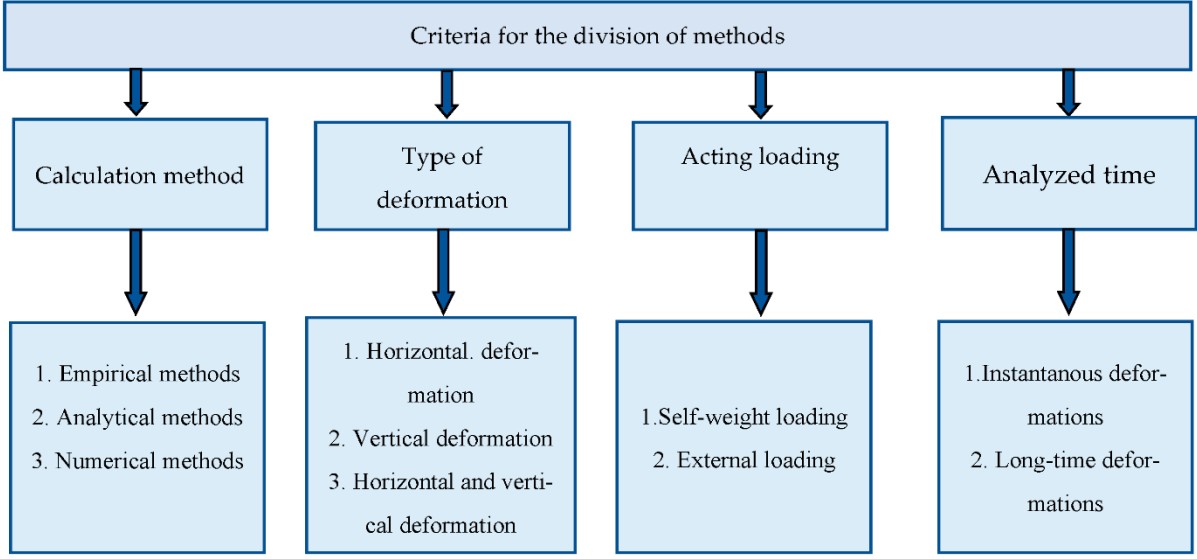

**Figure 2.** Methods used to investigate deformations in RS walls.

The second approach utilizes numerical methods to investigate the scope and magnitude of the deformations. Such calculations allow the assignment of material parameters that would be difficult to recognize in experimental studies. It should also be noted that the development of typical numerical procedures led to some important idealizations of the problem with regard to the geometry of the RS structure, the type of loading, the properties of the materials, and the boundary conditions. Therefore, the correctness of the proposed models and assumptions should be verified against extensive experimental data.

The methods used to solve particular cases (engineering or scientific problems) should be carefully selected. A short overview of the main information regarding analytical and numerical methods is presented below.

### 2.2. Analytical Methods

The most popular methods used for calculating the horizontal displacements of typical reinforced soil structures are shown in Table 1. The solutions that are commonly used are not very new, with most having been developed in the 1990s. In the 1990s, the level of professional knowledge was different and more limited. All of the factors/parameters that strongly influence the behavior of RS structures were not considered. Typically, they neglect the effects of the time-dependent properties of the materials used as reinforcement. Thus, the results obtained using these methods are usually not very accurate [77]. More detailed information about these analytical methods is presented by Kazimierowicz-Frankowska and Kulczykowski [40].

**Table 1.** Short description of the most popular analytical methods used to calculate the horizontal deformations of reinforced soil walls.

| Reference | Basic Correlations | Assumptions | Notations |
|---|---|---|---|
| [81] | $\delta_h = \frac{\varepsilon_d L}{2}$ | $\delta_h$—horizontal deformation of RS walls<br>$\varepsilon_d$—assumed strain limit | the strain limit is established as less than 10% |
| [73] | $\Delta_h = \left(\frac{1}{2}\right)\left(\frac{P_{rm}}{K_{reinf}}\right)(H - z_i) \cdot$ <br> $\cdot \left[\tan\left(45^0 - \frac{\psi}{2}\right) + \tan\left(90^0 - \phi_{ds}\right)\right]$ | $\Delta_h$—deformations of RS wall<br>$P_{rm}$—max. force<br>$\Psi$—dilatation angle of soil used as backfill | $\frac{L}{H} \geq 0.7$; |
| [82] | $D_L = \frac{2b_{q,vol} D_v}{H}$ <br> $\varepsilon_L = \frac{D_L}{b_{q,vol}} = \frac{2D_v}{H} = 2\varepsilon_v$ | $D_L$—horizontal deformation<br>$D_V$—vertical deformation<br>$\varepsilon_L$—horizontal strain<br>$\varepsilon_v$—vertical strain | the same values of soil and reinforcement horizontal deformations are assumed |

### 2.3. Numerical Methods

Numerical modeling has been successfully used by many researchers to evaluate the behavior of reinforced soil walls [20,23,25,83–88]. Typically, numerical approaches are divided into finite element methods (FEM) and finite difference methods (FDM). In order to perform numerical calculations, it is possible to use many different finite element numerical tools [89]. The most popular are DYNA3D, Plaxis, FLAC, Abaqus, and Ansys.

Table 2 presents information about the typical assumptions made in the numerical modeling of the behavior of RS structures.

**Table 2.** Typical assumptions in the numerical modeling of the behavior of RS walls.

| Reference | Code | Facing Model | Reinforcing Model | Soil Model | Soil/Reinforcement Interfaces | Soil/Wall Interfaces |
|---|---|---|---|---|---|---|
| [43] | FLAC | linear elastic element | elastic-plastic (two-node cable element) | Mohr–Coulomb with hyperbolic stress–strain model | grout material with zero thickness | slip element |
| [90] | Abaqus | linear elastic manner; eight-node plane strain elements | linear elastic manner; three-node truss elements having no significant compressive or bending strength | Mohr–Coulomb failure criterion and non-associated flow | grout material with zero thickness | slip element |
| [91] | PLAXIS | linear elastic element | linear elastic geogrid element | stress-dependent hyperbolic Hardening Soil (HS) model | grout material with zero thickness | slip element |
| [88] | FLAC | linear elastic element | linearly elastic-plastic strip element | Cap-Yield (CY) soil model | linearly elastic-perfectly plastic model with the Mohr–Coulomb failure criterion | linearly elastic-perfectly plastic model with the Mohr–Coulomb failure criterion |
| [92] | ABAQUS | linearly elastic element | 1D bar element, elastoplastic viscoplastic bounding surface model | Drucker–Prager creep model modified with nonlinear and cyclic hysteric behavior | thin layer elements follow Mohr–Coulomb failure criterion | thin layer elements follow Mohr–Coulomb failure criterion |
| [89] | FLAC3D | linearly elastic element | three-node shell elements; isotropic linear-elastic material | Mohr–Coulomb with hyperbolic stress–strain model | linear spring-slider system | linear spring-slider system |
| [93] | PLAXIS | three-node beam elements with flexural rigidity and normal stiffness | elastic material | elastoplastic Mohr–Coulomb material | Interface elements with three pairs of nodes, characterized by zero thickness | Interface elements with three pairs of nodes, characterized by zero thickness |

## 3. Laboratory Model Tests

### 3.1. Experimental Set-Up

The vertical retaining wall of reinforced soil was constructed and loaded at laboratory conditions. The experiment was carried out in a rigid test box with inside dimensions of 100 cm (height) × 37 cm (width) × 180 cm (length). The front and back walls of the box were made of glass (both 2 cm thick) in order to observe deformations of the model during loading. These walls were not lubricated, ensuring that good images of the soil deformations could be captured. The RS model consisted of 10 layers of sand with fabric reinforcement laid between them. The model was characterized by the following dimensions: 50 cm (height) × 60 cm (length) × 37 cm (width). The model of the footing used for the test was made of a 2.5 cm thick rigid steel plate with a rough base of 29 cm × 37 cm. The length of the plate was equal to the width of the tank to maintain plane strain conditions. The footing was connected to a vertical pull-out loading arm with a load cell, which in turn was

connected to an electro-mechanical actuator. The footing displacement was measured by displacement transducers. Five horizontal displacement transducers were located along the height of the RS model's facing in order to measure lateral displacements during loading. A schematic view of the test configuration is presented in Figure 3.

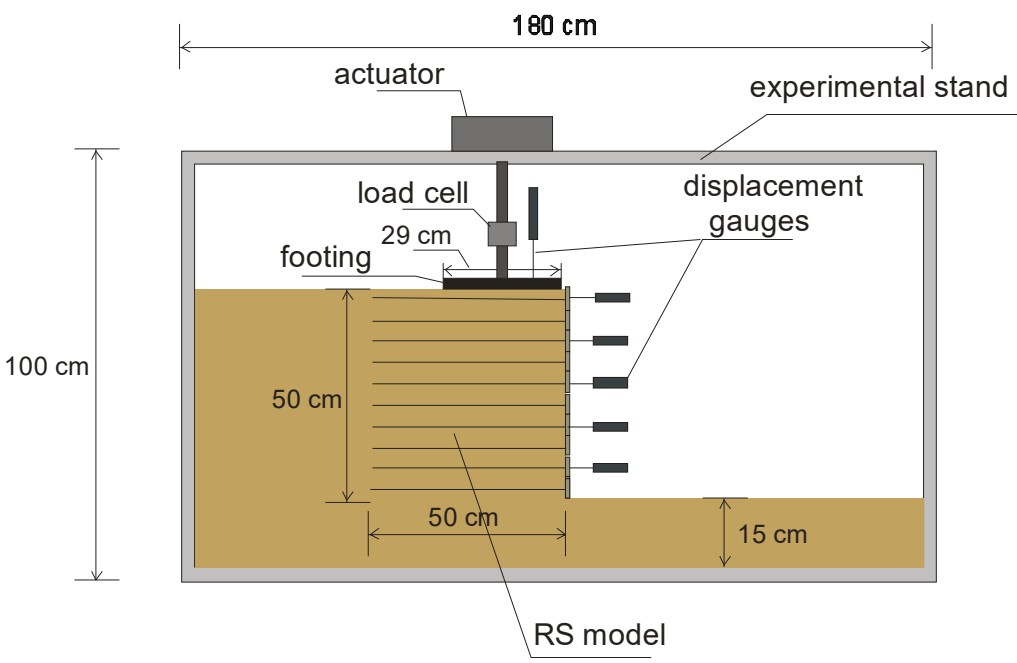

**Figure 3.** Cross-section of the experimental stand showing the positions of transducers.

*3.2. Characteristics of Materials Used in the Experimental Procedure*

3.2.1. Backfill

Dry silicon sand was selected as the backfill soil in the experimental stand. A sieve analysis was performed in order to determine the particle size distribution curve. Based on the results obtained, the grain size distribution curve of the material used as backfill was plotted (Figure 4). The coefficient of uniformity was 2.0 and the average particle size was 0.15 mm. The unit weight was $\gamma = 18.5 \times 10^3$ N/m$^3$, the relative density = 0.73, and the void ratio = 0.47 (Table 3). The friction angle of the sand obtained from standard triaxial compression tests was $\phi = 34.5°$ and the cohesion of the soil was $c = 0$ kPa.

**Table 3.** Properties of the main materials used for construction of the RS wall.

| Parameters | Unit | Value |
|---|---|---|
| Soil: sand | | |
| Unit weight $\gamma$ | kN/m$^3$ | 18.5 |
| Friction Angle $\phi$ | degrees | 34.5 |
| Cohesion $c$ | kN/m$^2$ | 0 |
| Relative density | - | 0.73 |
| Void ratio | - | 0.47 |
| Average particle size | mm | 0.15 |
| Uniformity coefficient | - | 2.0 |
| Reinforcement: aluminum foil | | |
| thickness | m | $18 \times 10^{-6}$ |
| ultimate tensile strength R | N/m$^2$ | $61 \times 10^6$ |
| stiffness $E$ | N/m$^2$ | $3533 \times 10^6$ |
| elongation at failure | % | 3.7 |
| coefficient of friction between the soil and the reinforcement | - | 0.05 |

**Table 3.** *Cont.*

| Parameters | Unit | Value |
|---|---|---|
| Parameters of RS model | | |
| height of the structure $H$ | m | 0.5 |
| length of the reinforcement strips $L$ | m | 0.5 |
| vertical spacing of the reinforcement $\Delta H$ | m | 0.05 |
| horizontal spacing of the reinforcement $\Delta B$ | m | 0.123 |
| width of a single reinforcement strip $B$ | m | 0.01 |

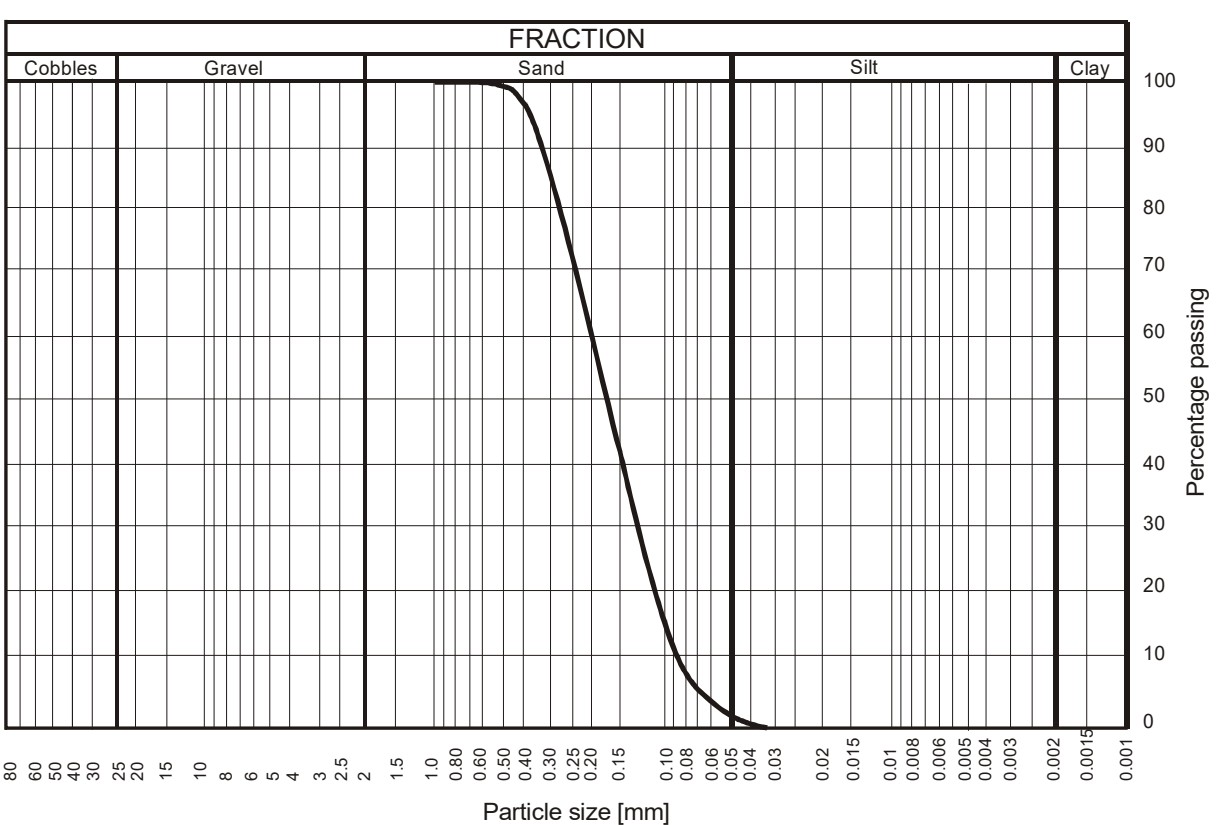

**Figure 4.** Grain size distribution curve of the material used as backfill.

### 3.2.2. Reinforcement

Aluminum foil with a thickness of $18 \times 10^{-6}$ m was used as the reinforcement. The load–elongation behavior of this foil was determined from a standard tension test (Figure 5).

The ultimate tensile strength measured was R = $61 \times 10^6$ N/m². Reinforcement stiffness reached the value of E = $3533 \times 10^6$ N/m² and elongation at failure was equal to 3.7%. The coefficient of friction between the backfill and the reinforcement, calculated from a direct shear test, was 0.18. The model reinforcement in each layer consisted of three independent fabric strips (5.0 cm wide and 50 cm long) with a center-to-center spacing of 12.3 cm (Figure 6b).

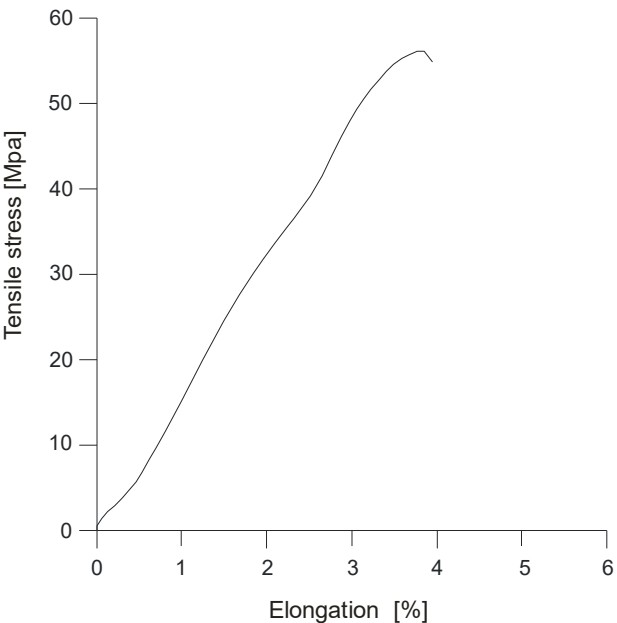

**Figure 5.** Stress-elongation behavior of the material used as reinforcement.

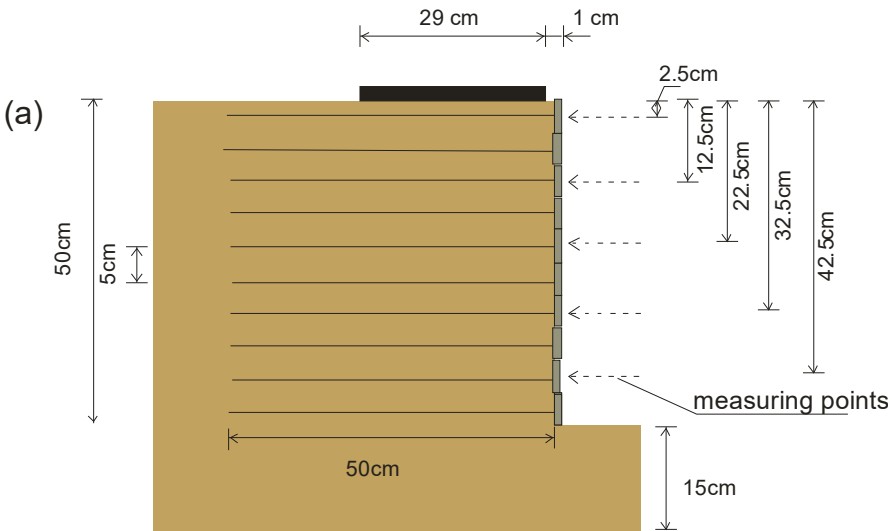

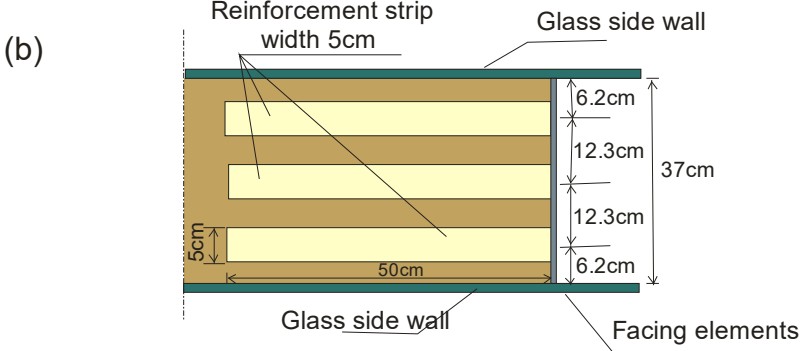

**Figure 6.** Cross-section of the RS model (**a**); layout of reinforcing strips (**b**).

### 3.2.3. Facing

The model reinforcement was connected to the elements used as wall facings. The timber panels were 5 cm high, 37 cm long, and 0.7 cm thick. The panels were made of pine wood with a unit weight of $\gamma_w = 4$ kN/m$^3$, a Young's modulus of $E_w = 12 \times 10^6$ kN/m$^2$, and a Poisson ratio of $\nu_w = 0.05$.

### 3.3. Testing Procedure

The sand was placed into the test box using pluviation technique. The height of fall was 100 cm to obtained the assumed unit weight of the backfill ($\gamma = 18.5 \times 10^3$ N/m$^3$). First, a 15 cm layer of soil foundation was formed at the bottom of test box. Then, a temporary support for the model, in the form of a wooden element, was positioned in front of the wall face in order to keep the facing surface in its initial place during construction. The model was built layer by layer from the bottom to the top, with 5 cm spacing between the soil and reinforcement layers. Each layer of soil was flattened with a grader. After this, a layer of reinforcement (consisting of three strips connected to the facing element) was placed on the sand, and then another sand layer was set down. This procedure was repeated until the structure was complete. Next, after the temporary support was removed, five displacement transducers were fixed on the wall facing at 5 points with regular intervals (Figure 6a). The actuator with the load cell and footing was then placed over the top of the model. The displacement transducers and the load cell were connected to a digital data logger. Finally, loading was applied on top of the structure through a footing that was pushed downwards at a constant displacement rate of 0.22 cm/min. The model was loaded continuously until failure was achieved with continuous measurements of the loading force, footing displacement, and facing deformation. The strip footings were loaded at a rate equal to those employed by other authors [94].

### 3.4. Experimental Results

The model failed rapidly at an applied vertical load of 3450 N and a vertical footing displacement of 0.5 cm. The relationship between vertical load and footing displacement is presented in Figure 7.

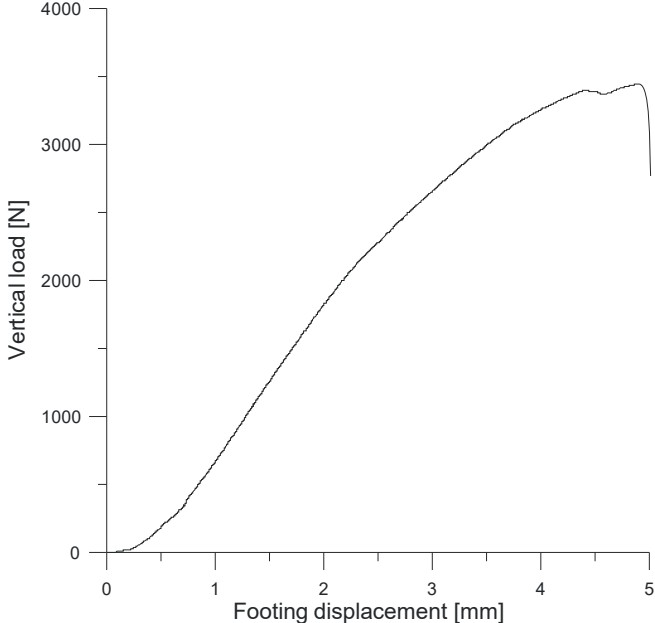

**Figure 7.** Load vs. footing displacement.

Figure 8 presents horizontal displacements of the facing measured at different loading levels. The same tendency was observed for all cases. The largest values of displacement

were measured in the top parts of the wall and diminished as the height of the structure decreased. The minimum horizontal movement was measured at the base of the structure. The measured values of deformations increased with increasing values of external loading.

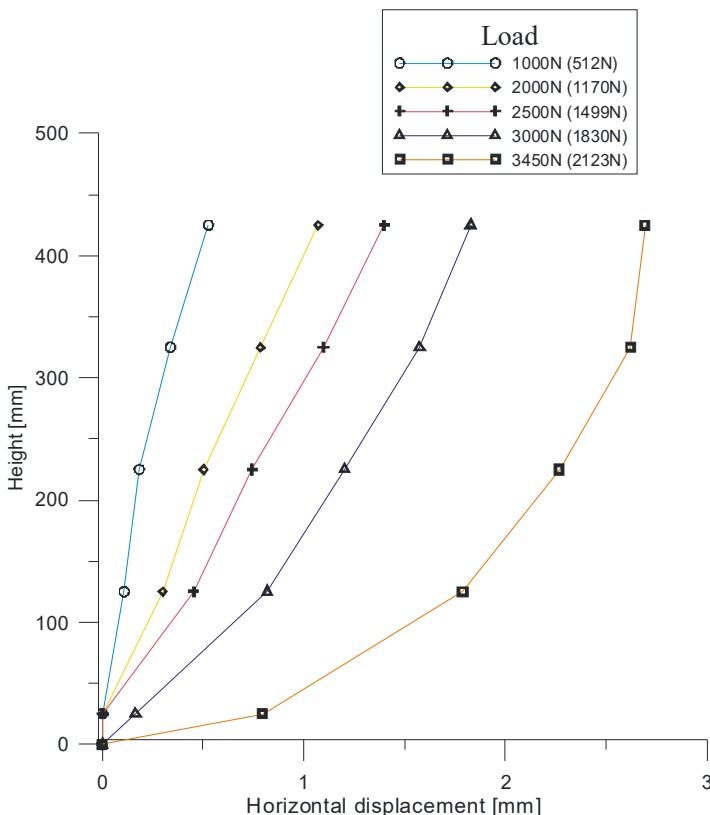

**Figure 8.** Horizontal deformation of the facing recorded at different loading levels.

The failure surface of the structure determined by the location of the points of tensile rupture of the reinforcement in each layer is presented in Figure 9. The theoretical Rankine failure surface (inclined to the horizontal plane at $\pi/4 + \phi/2$) is also plotted in this figure using a dotted line. It should be noted that the experimental slip surface corresponded reasonably well to this theoretical surface, and the experimental failure zone resembled a triangular wedge.

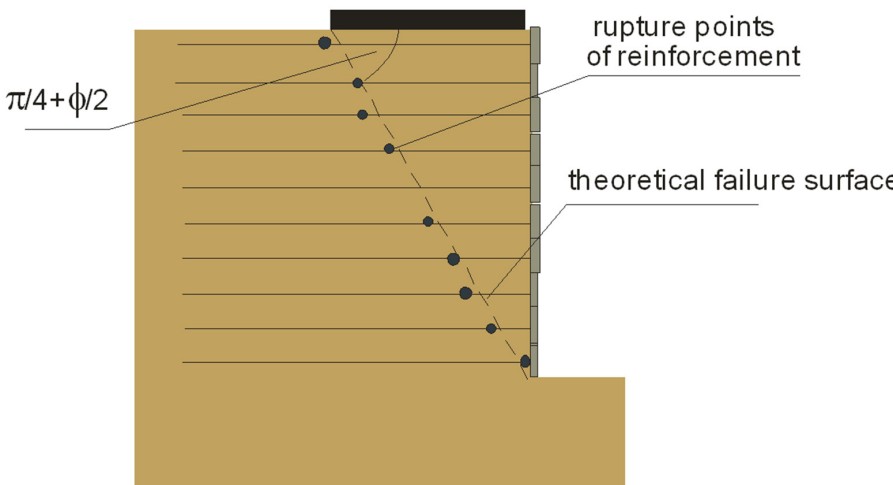

**Figure 9.** Experimental and theoretical Rankine failure surfaces.

### 3.5. The Effect of Friction between the Backfill and Side Walls of the Test Box

As previously noted, the front and back walls of the box were not lubricated. Therefore, the effect of friction between the backfill and the side walls on the load force should be taken into account. In order to estimate the increase in load due to such frictional resistance, a simple approach proposed by Kulczykowski [95] was used.

The resistance force ($P_F$) caused by the side wall friction can be estimated using the following formula, see [95]:

$$P_F = a^2 \tan \alpha \left[ \tan \delta (1 - \sin \phi) \left( \frac{a\gamma \tan \alpha}{3} + \frac{P_{E+F}}{aL} \right) + c_\delta \right], \tag{1}$$

where

$$a = \frac{H_P}{\tan \alpha}, \tag{2}$$

$\alpha$ is the angle of inclination of the slip surface to the horizontal plane, $\phi$ is the friction angle of the sand, $\delta$ is the angle of friction between the backfill and the side wall surface, and $c_\delta$ is the resistance resulting from adhesion (in N/m$^2$). $P_{E+F}$ is the load force recorded during the experiment (in N), $H_E$ is the height of the failure zone, and $\Delta L$ denotes the width of the test box (both in m) (Figure 10).

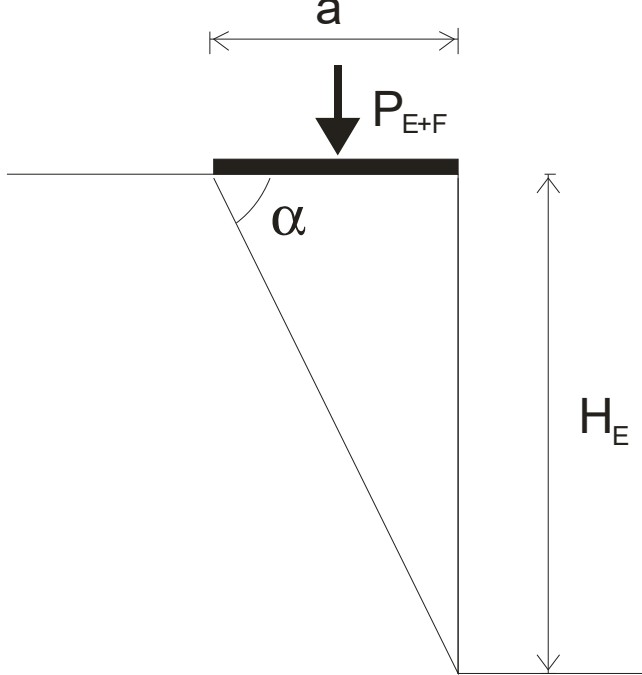

**Figure 10.** Triangular contact area between the sand and the side wall in the failure zone.

The value of the experimental load ($P_E$) reduced by the friction effect can be calculated using the following equation:

$$P_E = P_{E+F} \left[ 1 - \frac{a}{\Delta L} \tan \delta \tan \alpha (1 - \sin \phi) \right] - a^2 \tan \alpha \left[ \frac{1}{3} (1 - \sin \phi) a\gamma \tan \delta \tan \alpha + c_\delta \right] \tag{3}$$

The value of $\delta$ was determined by the method presented in [96] (a "tipping experiment"). A layer of sand was placed on the surface of sheet glass, which was the same as that used for the side walls of the test box. Then the sheet glass was slowly tilted. The angle of tipping was gradually increased until the soil mass began to slide. This angle of inclination was taken as the angle of friction $\delta$ between the sand and the glass surface.

The analysis of the digital images captured during a similar test and processed by particle image velocimetry (PIV), presented in [96], showed that a deformation zone with a

shape similar to that of the failure zone formed even at an early loading stage. Therefore, it was assumed that the above relationship can be used to calculate the value of $P_E$ at any stage of loading.

The following data, corresponding to the experimental conditions, were used to calculate the value of the load ($P_E$) at seven different stages of experimental loading: $P_{E+F}$: $H_E = 0.5$ m, $\phi = 34.5°$, $\gamma = 18.5 \times 10^3$ N/m³, $\delta = 26°$, and $c_\delta = 0$ N/m². The results of these calculations are presented in Table 4.

**Table 4.** Recorded reduced external load.

| $P_{E+F}$ [N] | $P_E$ [N] |
|---|---|
| 500 | 184 |
| 1000 | 512 |
| 1500 | 841 |
| 2000 | 1170 |
| 2500 | 1499 |
| 3000 | 1830 |
| 3450 | 2123 |

## 4. Theoretical Analysis: Analytical Approach

### 4.1. Basic Assumptions

The following initial assumptions were made [18,40,50]:

- The typical cross-section of the RS retaining wall was taken into consideration (Figure 11).

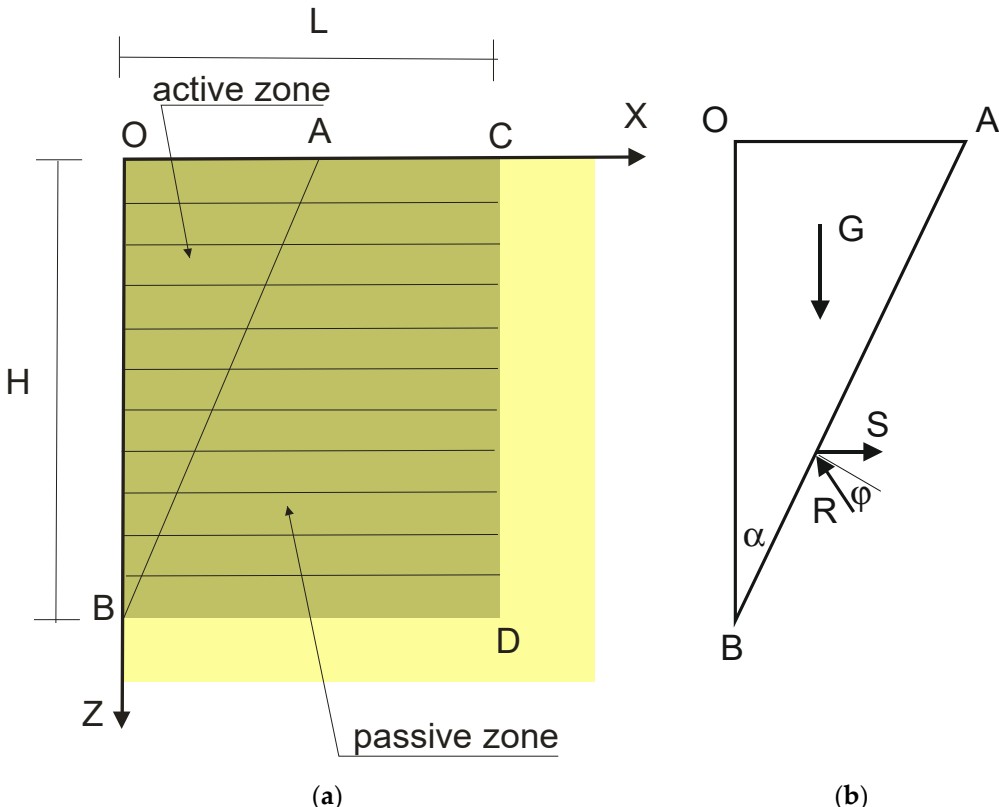

(**a**)                                                    (**b**)

**Figure 11.** General model of the RS wall (**a**) and equilibrium of the failure wedge (**b**).

- The soil was in a plastic state within the potential failure wedge *ABO* (active zone) and was rigid outside this area (passive zone).

- Perfect bonding between the soil and the reinforcement was assumed (slippage of the reinforcement did not occur).
- The increasing external load (acting on the top of the structure) was, at first, smaller than, and eventually equal to, the collapse load.
- The slippage of the wedge occurred along a planar failure surface that passed through the toe of the structure.
- The horizontal displacement ($u_x$) of the facing of the model RS wall consisted of two parts:

$$u_x = u_{act} + u_{pass} \tag{4}$$

where $u_{act}$ is the displacement resulting from the deformation of the reinforcement in the active zone, and $u_{pass}$ is the displacement resulting from the deformation in the rigid (passive) zone (caused by pull-out, hence $u_{pass} = u_{pullout}$).

*4.2. Deformation in the Plastic Zone*

The following constitutive relationship between tensile force and strain can be assumed in the elastic range (Hooke's law):

$$F = E\varepsilon_{act} \tag{5}$$

where $F = A_r\sigma_X^r$ is the force in the reinforcement, $E = A_rE_r$ is the elastic stiffness of the reinforcement, and $A_r$ is the cross-sectional area of the reinforcement.

The horizontal displacement of the RS facing caused by the elastic deformation of the reinforcement in the active zone is determined by the integration of strains:

$$u_{act} = \int_0^{x^*} \varepsilon_{act}dx \tag{6}$$

where $x^*$ denotes the length of the reinforcement in the active zone (Figure 12)

$$x^* = (H - z)\tan\alpha \tag{7}$$

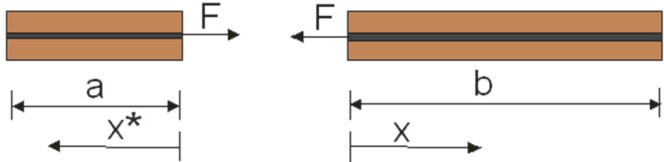

**Figure 12.** Overview of forces acting on the reinforcing element.

*4.3. Deformation in the Rigid Zone*

Displacements in this zone are caused by pull-out of the reinforcement. Sawicki [18], Kazimierowicz-Frankowska [50], and Kazimierowicz-Frankowska and Kulczykowski [40] presented detailed formulas which can be used to calculate these kinds of deformations.

For this purpose, the following governing equations can be used:

(1) The differential equation derived from the equilibrium of the reinforcing element (Figure 13) [18,50]:

$$\frac{dF}{dx} = -2B\tau, \tag{8}$$

where $F$ denotes the tensile strength of the reinforcement, and $B$ is the width of the reinforcement strip.

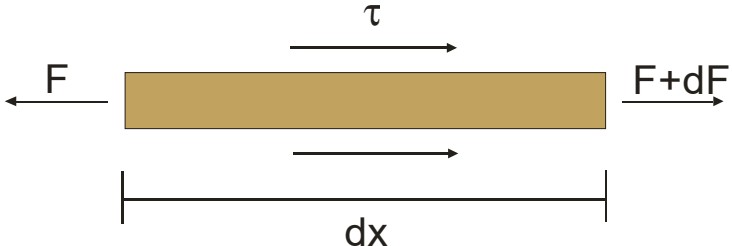

**Figure 13.** Reinforcing strip.

(2)    The equation describing strain in the reinforcement strip:

$$\varepsilon = du/dx, \tag{9}$$

where $u$ is displacement.

(3)    The correlation between sheer stress ($\tau$) and displacement ($u$):

$$\tau = -Gu, \tag{10}$$

where $G$ denotes a coefficient of proportionality.

(4)    The relationship between tensile strength and strain:

$$F = E\varepsilon, \tag{11}$$

where $E$ is the stiffness of the reinforcement strip.

Equations (8)–(11) give rise to the differential equation, which shows the distribution of forces along the reinforcement strip:

$$\frac{d^2F}{dx^2} - \beta^2 F = 0 \tag{12}$$

where $\beta = \sqrt{\frac{2BG}{E}}$. The solution for Equation (8) is the following function:

$$F = C_1 e^{-\beta x} + C_2 e^{\beta x}, \tag{13}$$

where $C_1$ and $C_2$ are coefficients determined from the corresponding boundary conditions:

$$F(x = 0) = F \text{ and } F(x = b) = 0 \tag{14}$$

It is possible to estimate the displacement ($u_{pullout}$) of the reinforcement strip using Equations (8), (10), (13), and (14).

$$u_{pullout} = -\frac{F}{\beta E[1 - \exp(-2\beta b)]} \cdot [\exp(-\beta x) + \exp(-2\beta b)\exp(\beta x)] \tag{15}$$

where $b$ is the length of the reinforcement strip in the rigid zone ($b$ = L − $a$), compared to Figures 11 and 12. The coefficient $G$ (Equation (10)) can be calculated using $\beta = \sqrt{\frac{2BG}{E}}$. $\beta$ can be determined using the following equation:

$$C = -\frac{\beta E[1 - \exp(-2\beta b)]}{1 + \exp(-2\beta b)} \tag{16}$$

## 5. Comparison of Experimental and Theoretical Results

*Verification of the Accurate Prediction of Experimental Results*

Firstly, the experimental values of the horizontal displacements were compared with their analytical predictions. The experimental results took into account the effect of friction between the backfill and the side walls of the test box. It should be also noted that the horizontal displacement caused only by external loading was measured during the experiment.

The results obtained are listed in Table 5 and presented in Figure 14. It can be seen that the experimentally recorded horizontal displacements are lower than the predicted horizontal displacements. The largest differences between the recorded and calculated values of deformation were observed at the top of the model. This was due to friction between the sand and the lower surface of the footing, which significantly reduced the horizontal deformation in this zone. However, compatibility between the experimental results and their theoretical predictions was considered satisfactory (i.e., acceptable for engineering practice). The average difference between the measured and calculated values of horizontal deformation was not greater than twenty five percent. Differences between the model results and the realistic behavior of the RS model structure may result from errors and inaccuracies that arose when the key material parameters used for the construction of the model wall were determined, or over the course of the experiments.

**Table 5.** Calculated ($u_T$) and experimental ($u_E$) displacement of the facing at subsequent load levels.

| Dist. from the Top [m] | Displacement of the Facing at Subsequent Load Levels [mm] | | | | | | | | | | | | | |
|---|---|---|---|---|---|---|---|---|---|---|---|---|---|---|
| | 184 N | | 512 N | | 841 N | | 1170 N | | 1499 N | | 1830 N | | 2123 N | |
| | $u_T$ | $u_E$ | $u_T$ | $u_E$ | $u_T$ | $u_E$ | $u_T$ | $u_E$ | $u_T$ | $u_E$ | $u_T$ | $u_E$ | $u_T$ | $u_E$ |
| 0.025 | 0.31 | 0.25 | 0.86 | 0.53 | 1.42 | 0.79 | 1.97 | 1.07 | 2.53 | 1.40 | 3.09 | 1.83 | 3.58 | 2.69 |
| 0.125 | 0.25 | 0.14 | 0.70 | 0.34 | 1.15 | 0.54 | 1.61 | 0.79 | 2.06 | 1.10 | 2.51 | 1.58 | 2.91 | 2.62 |
| 0.225 | 0.19 | 0.05 | 0.54 | 0.18 | 0.89 | 0.32 | 1.24 | 0.50 | 1.59 | 0.74 | 1.94 | 1.20 | 2.25 | 2.27 |
| 0.325 | 0.14 | 0.03 | 0.38 | 0.11 | 0.62 | 0.19 | 0.87 | 0.30 | 1.11 | 0.45 | 1.36 | 0.82 | 1.58 | 1.79 |
| 0.425 | 0.08 | 0.00 | 0.22 | 0.00 | 0.36 | 0.00 | 0.50 | 0.00 | 0.64 | 0.00 | 0.78 | 0.17 | 0.91 | 0.80 |

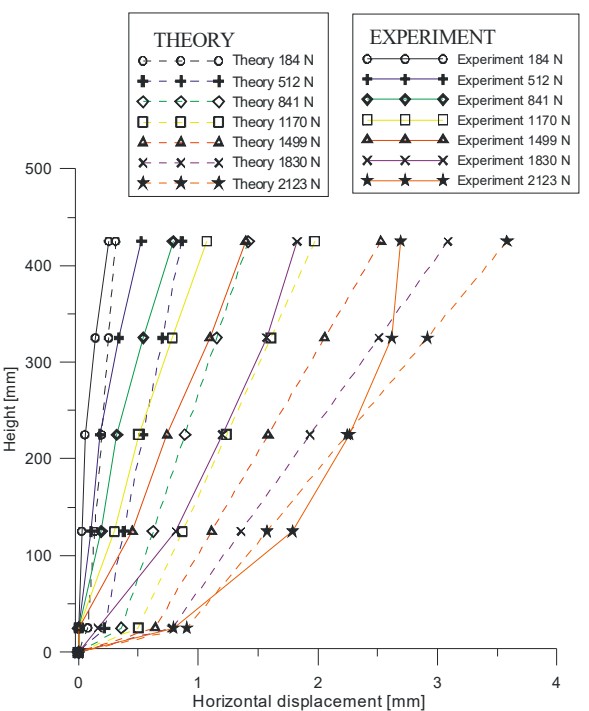

**Figure 14.** Comparison of experimental results with their analytical predictions.

The predicted horizontal displacements were larger than those that were experimentally recorded. From the point of view of design standards, there is a tendency for this to occur. Our results confirm that the proposed analytical method upholds the principles of safe design, which includes safety factors related to the potential occurrence of unexpected events.

## 6. Conclusions and Discussion

This paper aimed to investigate the accuracy with which horizontal deformations in the RS wall are modelled. This was achieved by comparing predictions of horizontal deformations with the experimental results obtained after constructing an instrumented RS wall on a laboratory scale. The displacements were recorded under external loading on top of the structure, and the theoretical analysis was carried out using an analytical approach.

The main findings of this study are as follows:

- The experimental investigations of the reinforced soil wall made it possible to observe the magnitude and pattern of horizontal displacements under a sustained external load at the top of the model. The largest displacements occurred in the top layers of the RS structure and gradually decreased as the height of the model wall decreased. Our results were similar to those observed by other researchers [40,50,51].
- An important and new insight that this paper introduces is the effect of friction between the backfill and the side walls of the test box on displacement. The effects of this factor tend to be omitted, or only generally discussed, by the authors of other experiments. However, our results show that friction has a significant impact on the value of external loading actually acting on the structure. In the analyzed case, even with the highest value of external loading on top of the structure, minimal reductions in the value of external loading actually acting on the structure were above 60% (Table 4). For example, the minimal reduction was 62.5% for an external load of 3450 N, 63.9% for an external load of 3000 N, and 66.8% for an external load of 2500 N. Therefore, the results clearly show that friction between the backfill and the side walls of the test box significantly influenced the horizontal displacement of the RS wall during the tests.
- Our results suggest that the proposed analytical method can be used as an alternative approach to other analytical and numerical methods used to model the deformations of RS wall structures. Although far from exhaustive, the first verification of the accuracy of the IBW PAN method produced promising results.
- The average difference between the recorded and calculated values of deformation did not exceed twenty five percent. Discrepancies between the model predictions and the experimental results may have resulted from inaccurate soil and reinforcement data.
- The theoretical horizontal displacements that were analytically predicted for the model wall were larger than the experimental horizontal displacements. This tendency is correct and shows that the proposed analytical method can be used without breaching safety rules.
- More complex constitutive soil models are investigated in the literature, but these models require input properties that are seldom available to design engineers. Moreover, greater accuracy in the predictions for more advanced models may not be assured [92]. Considering the accuracy of the results obtained, it should be stated that the proposed approach (and the models selected) are adequate from an engineering point of view.
- The final results may be particularly useful when designing the reinforced soil structures used in transport engineering (e.g., bridge abutments), where the deformation limit values cannot be exceeded. In such cases, the key factor determining the permissible durability of the structures is the second limit state of use (related to the size of the deformations occurring).

**Author Contributions:** Conceptualization, K.K.-F. and M.K.; Investigation, M.K.; Methodology, K.K.-F.; Resources, K.K.-F.; Supervision, K.K.-F.; Visualization, M.K.; Writing—original draft, K.K.-F.; Writing—review & editing, M.K. All authors have read and agreed to the published version of the manuscript.

**Funding:** This research received no external funding.

**Institutional Review Board Statement:** Not applicable.

**Informed Consent Statement:** Not applicable.

**Data Availability Statement:** The data presented in this study are available on request from the authors.

**Conflicts of Interest:** The authors declare no conflict of interest.

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
