# Peer review of "Laboratory Testing and Theoretical Modeling of Deformations of Reinforced Soil Wall"

_applsci, doi:10.3390/app12146895_

Round 1

Reviewer 1 Report

1.English writing of the whole manuscript should be improved. eg. “with showing”, “to construction”, ”They decreased down the height of the model wall.”, “is also may be useful”, 

2.Why the relative density is just 0.73?

3.Was the actuator used in this experimental study?

4.The big difference between experimental and calculation vaule under the condition of Experiment 2123 N should be analyzed.

5.The significance and or new contribution of this research work should be considered.

Author Response

Answer the Reviewer's questions/comments is presented in the enclosed attachment.

Reviewer 2 Report

This manuscript describes the construction of an instrumented RS wall on a laboratory scale, as well as the recorded measurements and analytical modeling of horizontal deformations in that wall. The manuscript may provide theoretical support for the construction of the RS wall. In general, the paper is well written and provide abundant information. It can be accepted after addressing the following questions:

1. The Abstract and Conclusion should be further improved to include not only the qualitative evaluation but also the quantitative analysis.

2. Too references are cited in the manuscript, however, few papers are published in recent five years. The references that are not relevant to the research should be removed, and some recently published papers can be added.

Author Response

Reviewer 2

Re:  Manuscript applsci-1721538 “Laboratory Testing and Theoretical Modeling of Deformations of Reinforced Soil Wall” by K. Kazimierowicz-Frankowska and. M. Kulczykowski submitted to Applied Sciences (special issue titled: “New Frontiers in Sustainable Geotechnics”).

Reply to comments: Reviewer 2

We would like to start by thanking the Reviewer for the valuable and helpful comments which have allowed us to improve the quality of the article and make it more accessible to the readers.

The original comments:

  1. The original Comment 1:

“1. The Abstract and Conclusion should be further improved to include not only the qualitative evaluation but also the quantitative analysis”.

1.1   Reply to Comment 1:

The Abstract and Conclusion parts of the paper have been changed. The content of the abstract has been changed/supplemented in accordance with the Reviewer's comments. The information regarding significance and new contribution of this research work has been added. However, given the word limit (“The abstract should be a total of about 200 words maximum”), this data is presented in a summary. There was also a slight reduction of some of the text originally posted to make space for the new information. In addition, the new version of the abstract tried to take into account the comments of all other Reviewers, who indicated the following: “The abstract needs improvement by adding justification of the research, some fundamental data from results that show what was found and the field application of the research findings should be captured in the abstract”. 

  1. The original Comment 2:

“2. Too references are cited in the manuscript, however, few papers are published in recent five years. The references that are not relevant to the research should be removed, and some recently published papers can be added”.

2.1  Reply to Comment 2:

The list of references has been changed according to the Reviewer’s suggestions. The following positions (recently published papers) have been added to the list of references:

  1. Gawali, S.L, L. G. Kalurkar, L.G. Deformation Behaviour of Geogrid Reinforced Soil Walls under Strip Loading, International Journal of Engineering Research & Technology (IJERT), 9 (1), 2020, 23-30.
  2. Riccio, M., Ehrlich, M. Engineered Fabrics,  Edited by Mukesh Kumar Singh, 2018.
  3. Lu, L., Lin, H., Wang, Z., Xiao, L., Ma, S., Arai, K. Experimental and Numerical Investigations of Reinforced Soil Wall Subjected to Impact Loading, Rock Mechanics and Rock Engineering, 54, 2021, 5651–5666.
  4. Jia, X., Xu, J., Sun, Y. Deformation Analysis of Reinforced Retaining Wall Using Separate Finite Element, Discrete Dynamics in Nature and Society (DDNS), 2018,  https://doi.org/10.1155/2018/6946492
  5. Wu, J. T. H. Geosynthetic Reinforced Soil (GRS) Walls, Wiley Blackwell, 2nd Edition, 2020
  6. Yoo, C., Tabish, A., Yang, J.W., Abbas, Q., Song, J.S. Effect of internal drainage on deformation behavior of GRS wall during rainfall, Geosynthetics International, 29 (2), 2022,  137-150.

If the Reviewer believes that the list of references requires further changes, we would be grateful for if the Reviewing could give us further suggestions. In the current version, none of the items originally posted have been removed (other Reviewers have not raised any objections to the bibliography), but we are open to further comments and suggestions and will try to include them in the final version of our paper.

Reviewer 3 Report

Vertical deformation vertical in Fig 2. What does it mean?

Author Response

Re:  Manuscript applsci-1721538 “Laboratory Testing and Theoretical Modeling of Deformations of Reinforced Soil Wall” by K. Kazimierowicz-Frankowska and. M. Kulczykowski submitted to Applied Sciences (special issue titled: “New Frontiers in Sustainable Geotechnics”).

Reply to comments: Reviewer 3

We would like to start by thanking the Reviewer for the valuable and helpful comments which have allowed us to improve the quality of the article and make it more accessible to the readers.

The original comments:

  1. The original Comment 1:

“Vertical deformation vertical in Fig 2. What does it mean?”.

1.1   Reply to Comment 1:

A previous error in the description of Figure 2 has been corrected. Figure 2 shows the classification of methods used for analysis of the RS structures’ deformations. It is possible to select a few main approaches to this problem. In one of them the direction of acting deformation has been taking into account as the main criterium for the division of these methods.

As part of the description of this criterion, the following 3 points should be included:

  1. Horizontal deformation
  2. Vertical deformation
  3. Horizontal and vertical deformations

As the Reviewer rightly pointed out, in the original version of the article, the third point was described incorrectly.

Reviewer 4 Report

The paper Laboratory Testing and Theoretical Modeling of Deformations of Reinforced Soil Wall has presented a comparative study of numerical and analytical investigation. This is supported by laboratory examinations. 

The abstract needs improvement by adding justification of the research, some fundamental data from results that show what was found and the field application of the research findings should be captured in the abstract. 

There should be minor English language checks for language problems and possible typo errors.

The section comparing laboratory examinations and analytical studies results needs extensive information and improvement. 

The conclusion is sufficient 

The reference list and associated citations are according to journal format 

Author Response

Reviewer 4                                                                                                                                                      

Re:  Manuscript applsci-1721538 “Laboratory Testing and Theoretical Modeling of Deformations of Reinforced Soil Wall” by K. Kazimierowicz-Frankowska and. M. Kulczykowski submitted to Applied Sciences (special issue titled: “New Frontiers in Sustainable Geotechnics”).

Reply to comments: Reviewer 4

We would like to start by thanking the Reviewer for the valuable and helpful comments which have allowed us to improve the quality of the article and make it more accessible to the readers.

The original comments:

  1. The original Comment 1:

“The abstract needs improvement by adding justification of the research, some fundamental data from results that show what was found and the field application of the research findings should be captured in the abstract”. 

     1.1   Reply to Comment 1:

The content of the abstract has been changed/supplemented in accordance with the Reviewer's comments. The information regarding significance and new contribution of this research work has been added. However, given the word limit (“The abstract should be a total of about 200 words maximum”), this data is presented in a summary. There was also a slight reduction of some of the text originally posted to allow for the additional information to be included.

  1. The original Comment 2:

“There should be minor English language checks for language problems and possible typo errors”.

    2.1   Reply to Comment 2:

The authors corrected the noted language errors.  Especially chapters such as: “Introduction” and “Conclusions and discussion” have been changed/improved. We are open to additional suggestions in this regard. If necessary, we will make additional changes.

  1. The original Comment 3:

“The section comparing laboratory examinations and analytical studies results needs extensive information and improvement”. 

3.1   Reply to Comment 3:

Description of the section comparing laboratory examinations and analytical studies’ results has been changed according to the Reviewer’s suggestion. Supplemented information has been added. It deals with two main elements:

  1. Description and discussion of the obtained theoretical results – more detailed information has been added.
  2. Section 5.1  “Verification of accurate prediction of experimental results” has been rearranged and expanded. Table 5 with a comparison of calculated and experimental values of facing displacement at subsequent load levels has been added to the text.